# Toxic-Induced Nonthyroidal Illness Syndrome Induced by Acute Low-Dose Pesticides Exposure—Preliminary In Vivo Study

**DOI:** 10.3390/toxics10090511

**Published:** 2022-08-29

**Authors:** Cristian Cobilinschi, Radu Țincu, Raluca Ungureanu, Ioana Dumitru, Alexandru Băetu, Sebastian Isac, Claudia Oana Cobilinschi, Ioana Marina Grințescu, Liliana Mirea

**Affiliations:** 1Department of Anesthesiology and Intensive Care, Clinical Emergency Hospital of Bucharest, 014461 Bucharest, Romania; 2Department of Anesthesiology and Intensive Care II, Carol Davila University of Medicine and Pharmacy, 050474 Bucharest, Romania; 3Department of Toxicology, Carol Davila University of Medicine and Pharmacy, 050474 Bucharest, Romania; 4Department of Anesthesiology and Intensive Care I, Fundeni Clinical Institute, 022328 Bucharest, Romania; 5Department of Internal Medicine, Sf Maria Clinical Hospital, Carol Davila University of Medicine and Pharmacy, 050474 Bucharest, Romania

**Keywords:** thyroid impairment, sick euthyroid syndrome, organophosphate intoxication, TSH, T3, T4, pollutants, animal models, environmental health

## Abstract

Background and Objectives: Conditions such as trauma, burns, sepsis, or acute intoxications have considerable consequences on the endocrine status, causing “sick euthyroid syndrome”. Organophosphate exposure may induce an increase in acetylcholine levels, thus altering the thyroid’s hormonal status. The present study aims to identify the effects of acetylcholinesterase inhibition on thyroid hormones. Material and methods: A prospective experimental study was conducted on twenty Wistar rats. Blood samples were drawn to set baseline values for thyroid-stimulating hormone (TSH), triiodothyronine (T3), and thyroxine (T4). Chlorpyrifos 0.1 mg/kg was administered by oral gavage to induce acetyl-cholinesterase inhibition. After exhibiting cholinergic symptoms, blood samples were collected to assess levels of cholinesterase and thyroid hormones using ELISA. Results: Butyrylcholinesterase levels confirmed major inhibition immediately after intoxication compared to the baseline, certifying the intoxication. A significant increase in T4 levels was noted (*p* = 0.01) both at 2 h and 48 h after administration of organophosphate in sample rats. Similarly, T3 almost doubled its value 2 h after poisoning (4.2 ng/mL versus 2.5 ng/mL at baseline). Surprisingly, TSH displayed acute elevation with an afterward slow descending trend at 48 h (*p* = 0.1), reaching baseline value. Conclusions: This study demonstrated that cholinesterase inhibition caused major alterations in thyroid hormone levels, which may be characterized by a transient hypothyroidism status with an impact on survival prognosis.

## 1. Introduction

Acute medical conditions, such as trauma, sepsis, intoxication, and burns, cause alteration of the hypothalamic–pituitary–thyroid axis even since the initial phase [1]. Although not traditionally considered “stress hormones”, thyroid hormones participate in the acute phase of stress by maintaining vasomotor tone, thus influencing the hemodynamic status and myocardial inotropism [2]. 

“*Low T3 syndrome*”, originally characterized by Wartofsky and Burman, is a transient thyroidal dysfunction frequently identified in critically ill patients [3]. It has subsequently been shown that patients with this type of syndrome are clinically euthyroid, so the name has been changed to *“euthyroid impairment syndrome*” [4]. The term currently used for this type of impairment is “*non-thyroid impairment syndrome*” [4]. From a biochemical point of view, the most frequent change is a low triiodothyronine (T3) level, which occurs even in conditions of moderate systemic dysfunction [4]. In critical conditions, alterations of thyroxine (T4) and thyroid stimulating hormone (TSH) can also occur [4]. However, it has been observed that TSH values may be decreased, despite the establishment of clinical and biochemical hypothyroidism [5]. 

Available data in the literature indicate that the impaired ability of TSH to respond adequately to decreased T3 is suggestive of the onset of central hypothyroidism [4]. In addition, in parallel with the decrease in T3 values, there is an increase in the level of reverse T3 [6]. The rate at which changes in hormonal values occur is directly proportional to the severity of the underlying dysfunction [7]. Although these hormonal changes can be interpreted as an adaptation of the body aimed at reducing energy consumption in critical conditions, research has also found that low levels of thyroid hormones are correlated with an increased risk of mortality [7]. 

The pathogenic mechanism of this type of impairment is not fully known and is considered to be a concomitant involvement of altered synthesis, protein binding, cell uptake, deiodination, or receptor modulation [8]. 

Particularly in acute organophosphate (OP) exposure, increased cholinergic stimulation causes thyroid tissue damage [9]. It has been found that the accumulation of acetylcholine at the hypothalamic and pituitary levels stimulates the release of somatostatin, which leads to inhibition of the release of a thyroid-releasing hormone (TRH) and TSH, respectively [10]. In addition, oxidative stress caused by exposure to OP compounds and the release of inflammatory cytokines exacerbates thyroid dysfunction [6]. 

In the case of acute OP intoxication, as well as in the case of trauma or sepsis, the alteration of the hypothalamic–pituitary–thyroid axis is characterized by the appearance of a thyroid hypofunction without clear biochemical changes [11].

Beyond the clinical manifestations specific to the cholinergic toxidrome described since the synthesis of the first OP, special attention was paid to the neurotoxicity and to the impact on the neurodevelopment of these compounds [12]. Endocrine and especially thyroid damage have recently come to the fore, as there have been detected even in low-dose exposure conditions [13]. Recent experimental data also suggest that repeated exposure may cause severe endocrine disturbances in the absence of any typical cholinergic symptom [8].

Organophosphorus-type insecticides have now become the most widely used type of pesticides in the world [14], but the exposure to chlorpyrifos (one of the most used OP compounds in agriculture today) toxicity is not limited to agricultural populations but is extended to all environments given its use in controlling household insects or eliminating pet ectoparasites [15]. 

Considering data from the literature, we developed an experimental longitudinal study that aimed to evaluate changes in thyroid hormones after chlorpyrifos intoxication.

## 2. Materials and Methods

The experimental study included 20 adult Wistar rats purchased from an accredited biobase of the “Victor Babeș” National Institute for Research and Development. Considering the impact of the estrous cycle on hormonal secretion and behavior, only male rats were included in order to avoid any hormonal interferences [16]. All subjects included in the study were adults corresponding to an age of 60 days. The mean weight in the study group was between 320 and 406 g. 

All rats were initially tested for baseline butyrylcholinesterase, free-T3 (fT3), free-T4 (fT4), and TSH following blood sampling. These samples were taken by puncturing the tail vein with a 29 G needle. The sampling protocol followed the recommendations of the Institutional Committee for the Care and Use of Laboratory Animals (IACUC) for blood samples from laboratory animals, according to which the maximum amount of blood harvested within 24 h should not exceed 1% of body weight. The amount of blood needed to perform the established hormonal determinations did not require more than 1 mL of blood to be collected. During the entire study, the chlorpyrifos (ReldanTM 22 EC, Dow Agrosciences) was administered by means of an orogastric tube at a dose of 0.1 mg/kg, equivalent to half the lethal dose 50 (Chlorpyrifos LD50 for rats after oral exposure 0.2 g/kg according to World Health Organization Specifications and Evaluations for Public Health Pesticides –https://apps.who.int/iris/bitstream/handle/10665/332193/9789240005662-eng.pdf?ua=1 (accessed on 13 May 2022)).

The dilution of OP was performed with sterile saline, and the administration of the established dose according to body weight was performed in a mixture with corn oil to facilitate digestive absorption. At two hours and 48 h after OP administration, blood samples were taken according to the same protocol, considering that the aim of the study was to evaluate the impact on thyroid function in the hyperacute and acute phases of exposure. This study used a within-subjects design to evaluate the effects of OP exposure. In order to increase internal validity, some measures were taken to limit the impact of various extraneous factors which may interfere with the final results, such as sampling at the same time of the day and same light and heat exposure.

### 2.1. Animals and Housing

All laboratory animals selected for the study were fed with regular rodent chow food, with normal iodine content, purchased from the biobase of the “Victor Babeș” National Institute for Research and Development. The rats also benefited from water ad libitum. During the experiment, the night/day cycle was simulated with artificial light at 12-h intervals, and they were housed in an adequately ventilated room at a constant temperature of 22 °C. 

Each blood sampling, as well as the placement of the orogastric tube, was performed under general anesthesia according to the IACUC protocol for the anaesthesia of laboratory animals—the rats’ section, using 1% isoflurane and ketamine/xylazine 40/5 mg/kg.

### 2.2. Blood Sample Analysis

Blood samples were taken using vacuum collection tubes containing a coagulation activator. They were transported to a specialized laboratory where the preestablished determinations were performed. Butyrylcholinesterase levels were detected using a spectrophotometric method and were expressed in units/liter (U/L). The hormonal determinations set out in the study protocol were performed with the ELISA sandwich technique for quantitative determinations, using specific kits for rats. Thus, for T3, the kit with the identification code ABIN2685560 (Abnova, Taipei, Taiwan) was used; for T4, the identification code of the kit was ABIN365197 (Cusabio, Wuhan, China), and for TSH, it was ABIN2685829 (Abnova, Taipei, Taiwan). For all determinations, the values were expressed in nanograms/milliliter (ng/mL). 

### 2.3. Statistical Analysis

The obtained results were analyzed using MedCalc 14.1. (Ostend, Belgium) Data obtained following the use of statistical tests were expressed as average values ± standard deviation and standard error of mean. Considering the repeated-measurements crossover design of the study, baseline values for each subject were used as control values. As a result, data were expressed as ‘changes from baseline’. The comparative analysis of the means at three different checkpoints (baseline, 2 h, and 48 h after chlorpyrifos administration) was performed based on the analysis of variance test (ANOVA). In addition to assessing the significant differences between the determinations, the existence of possible correlations between the determined parameters was also assessed. Thus, based on the Pearson correlation coefficient (r), the proportionality relations between the results obtained from the study group were established. It was considered that a value of r between 0 and 1 corresponds to a proportional correlation, r between −1 and 0 indicates an inversely proportional relationship, and r = 0 reveals the absence of a correlation between the analyzed parameters. A supplementary correlation analysis using the scatter diagram function available in the MedCalc software was also performed. For all statistical tests applied in the study, a threshold of statistical significance was established if *p* ≤ 0.05.

## 3. Results

In the study group, after oral administration of chlorpyrifos, specific manifestations of the cholinergic syndrome were observed in three subjects (S3, S7, and S8). These clinical signs were observed in the first two hours after OP administration. S3 and S7 presented increased watery mucus in the nasal cavity and diarrhea, and S8 showed copious salivation, polyuria, and diarrhea. 

None of the mice were given the antidote, the remission of symptoms being spontaneous. None of the rats initially included in the study group died following the intoxication. The diagnosis of acute intoxication was made based on the biochemical determination of butyrylcholinesterase. This indicated a baseline value of 544 U/L ± 22.78 (SEM = 5.09), which then decreased in two hours to 0.59 U/L ± 0.43 (SEM = 0.09), and in 48 h increased to 282 U/L ± 47.2 (SEM = 10.5) (Figure 1). The differences between the three determinations were statistically significant (*p* = 0.0001).

Regarding the T3 level, it was observed that compared to the base level (2.48 ng/mL ± 0.34, SEM = 0.07), it increased at two hours to a value of 4.1 ng/mL ±0.5 (SEM = 0.12). Forty-eight hours after intoxication, its level had dropped to 3.1 ng/mL ± 1.05 (SEM 0.23). The differences between the three determinations were also statistically significant (*p* < 0.0001) (Figure 2).

The T4 value also increased from a baseline value obtained at the first determination of 34.95 ng/mL ± 3.8 (SEM = 0.8) to 43 ng/mL ± 5 (SEM = 1.12). Subsequently, unlike T3, the T4 level continued to increase 48 h after exposure to 44.3 ng/mL ± 12 (SEM = 2.7), as seen in Figure 3. The difference between the three measured values was again statistically significant (*p* = 0.0012). 

Regarding the TSH level, compared to the initial baseline value of 3.4 ng/mL ± 0.5 (SEM = 0.11), at the first determination it increased to 4.1 ng/mL ± 0.5 (SEM = 0.11), and later decreased to 3.5 ng/mL ± 0.5 (SEM = 0.11). The differences between the three determinations were statistically significant with a *p*-value of 0.0003; however, TSH values measured on 48 h, compared with baseline, were not statistically significant (Figure 4).

The calculation of the Pearson correlation coefficient revealed that the increase in the value of T4 and that of TSH at two hours post intoxication are inversely proportional (r = −0.51) and statistically significant (Figure 5). 

Another inversely proportional correlation (r = −0.399) was obtained between the T3 and TSH values measured two hours after the administration of chlorpyrifos (Figure 5). However, this correlation was close to reaching the threshold of statistical significance (*p* = 0.08). 

Regarding the determination of the hormonal level at 48 h, a weak correlation between T4 (r = 0.08), T3 (r = 0.24), and TSH was observed without any statistical significance.

No statistically significant correlation was identified using the scatter diagram available within the correlation analysis.

## 4. Discussion

Although they are considered to have a low persistence in the environment, the extensive use of OP insecticides has now led to their ubiquitous identification both in nature and in homes [13]. In the present trial, in the group of 20 adult Wistar rats, chlorpyrifos intoxication was induced orally by administering it through an orogastric tube. The absorption of chlorpyrifos administered by the digestive tract is thought to be rapid due to its unionized character and the increased fat solubility of the molecule [17]. Two hours after OP administration, three subjects presented mild muscarinic symptoms. 

Two hours after oral administration of chlorpyrifos, the level of butyrylcholinesterase was measured to establish the biochemical diagnosis of intoxication. Thus, significant inhibition of the butyrylcholinesterase level was observed compared to the baseline value, establishing a positive diagnosis of acute intoxication. The biochemical diagnosis of OP poisoning has traditionally been based on the measurement of blood levels of butyrylcholinesterase and acetylcholinesterase [18]. Despite acetylcholinesterase’s higher affinity to OP than butyrylcholinesterase, the very low amount of acetylcholinesterase in plasma set the diagnostic tests using acetylcholinesterase frequently out of detection range in OP exposure [19]. Butyrylcholinesterase is available in large amounts in the plasma, and diagnostic tests are widely available [20]. Eddleston et al. reported that butyrylcholinesterase activity detection immediately after chlorpyrifos exposure may be very useful, having a sensitivity of 100% [21]. Unlike the measurement of acetylcholinesterase, the use of butyrylcholinesterase levels also provides information on cases of exposure to low doses of OP, given its increased sensitivity [18]. However, butyrylcholinesterase may be influenced by other factors, such as physical exertion, weight, or age [22]. The persistence of inhibited butyrylcholinesterase indicates the persistence of intoxication, and its gradual return to normal levels correlates with the gradual elimination of OP compounds [18]. Forty-eight hours after intoxication, the level of butyrylcholinesterase in the study group showed an increase equivalent to half the baseline value. This indicates a persistence of the compound at the second determination as well. This result is supported by toxicokinetic studies indicating that thiophosphate (*p* = S) organophosphates, such as chlorpyrifos, have a higher fat solubility than phosphate (P = O) [17]. In other words, thiophosphates have an increased capacity to accumulate in the adipose tissue, prolonging the exposure period [17]. Following the inhibition of cholinesterase, as well as through the non-cholinergic mechanisms described more recently, it has been found that these compounds interfere with a variety of physiological processes, such as protein phosphorylation, cell development, modulation of immunity, or reproductive function [23]. In addition to changes in the regulation of sex hormones, disorders of thyroid hormones secondary to OP intoxication indicate their interference in the processes of development, maturation, and metabolism [24]. In the present research, two hours after intoxication, an increase in T3 levels was observed. This result coincides with existing data in the literature, according to which, under conditions of acute stress, the secretion of thyroid hormones increases [25]. Otenio et al. reported similar data in a very recent study that included only female rats [26]. However, most of the data on the evolution of T3 plasma levels in acute situations, including acute OP poisoning, indicate a rapid decrease in T3 within 18 h of admission [1,6,27]. A study conducted by Güven et al. found a decrease in T3 values in only two patients out of a total of 22 subjects [28]. Moreover, the differences between the initial measurements and those taken after the administration of the antidote did not meet the threshold for statistical significance [28]. Satar et al. reported in their experimental study that the T3 level decreased rapidly in the study group compared to the control group, with the onset of cholinergic manifestations [27]. In the current study, the determination at 48 h after intoxication, however, revealed a significant decrease in T3 level, which is consistent with existing data. 

Two hours after exposure, an increase in T4 values was observed, and measurements at 48 h indicated a continuing upward trend. These results are in line with those published in the literature. Slotkin et al. reported that T4 levels were also increased in intoxicated rats in the immediate postnatal period [29]. Contradictory data were obtained by Satar et al. research where, like the T3 level, the T4 value was decreased after metamidophos intoxication. One hypothesis of elevated T4 levels in the context of acute intoxication may be related to the impairment of T4 to T3 conversion by alteration of the deiodination process [1]. The evolution of the TSH level followed a similar pattern to the T3 values so that two hours after the administration of chlorpyrifos, TSH increased. Subsequently, at 48 h, its level dropped to a value close to the baseline. In the above-mentioned study conducted by Satar et al., it was observed that the level of TSH increased in the group of intoxicated rats that received antidote treatment [27]. Jeong et al. also observed an increase in the level of TSH under conditions of prolonged exposure to chlorpyrifos-methyl, both in females and males [30]. The significant decrease in TSH at 48 h can be explained by the exacerbation of the inhibitory effects of somatostatin due to the accumulation of acetylcholine resulting from the inhibition of acetylcholinesterase [28]. 

Statistical analysis of correlations revealed an inversely proportional relation between T4 and TSH values determined two hours after exposure to chlorpyrifos, which corresponds to the physiological mechanisms by which increased T4 causes negative feedback to decrease TSH release. A similar correlation was also obtained between T3 and TSH levels, measured two hours post-intoxication. At 48 h after exposure, the only statistically significant correlation was between T3 and T4, indicating a proportional increase in the two hormones. 

It has already been stated that non-thyroidal illness syndrome is related to a primary disease, and its severity is proportional to the latter. Wang et al. stated that the severity of this syndrome correlates to the APACHE II score, highlighting the importance of prompt intervention upon hormonal disbalance [31]. In the present study, the exposure level was not critical; however, significant hormonal changes were recorded in the study group. 

Leemans et al. showed that continuous exposure to low-dose OP might be responsible for the occurrence of thyroidal dysfunction, which may reflect on exposed children’s neurological development [12]. Cobilinschi et al. also reported that acute exposure to OP may induce significant hormonal disturbances in weanling rats, similar to low T3 syndrome [8]. Our study confirms that clinical signs after low-dose exposure are rarely identifiable, making it difficult to place a timely diagnosis and to incriminate OP poisoning for neurological disorders.

### Study Limitations

The main limitation of the present study is the absence of a control group, considering that it may be useful to evaluate the impact of extraneous variables and physiological parameters change over time, such as temperature or mobility. In the present study, baseline values were used as a type of control values, and data were expressed as ‘changes from baseline’. Considering that in the study group, the experiment was completed in a very short period of time and the study group was homogenous, the use of baseline values can be accepted as control values. Nevertheless, using baseline values allowed us to limit the included subjects and, at the same time, expand statistical power by using within-subject statistics [32]. 

Another limitation of this research may be the absence of a hormonal determination at a larger time interval from the moment of intoxication, which could have provided the information necessary to establish with certainty the characteristics of the type of damage from chlorpyrifos intoxication. However, the main purpose of this research, namely the assessment of endocrine dysfunction in the acute phase of intoxication, should not be overlooked. Testing the reverse T3 level would have also provided data on the amount of T3 that is inactivated under acute exposure conditions, given that a low T3/rT3 ratio is generally associated with an increased risk of mortality [1].

The exclusion of female subjects may also be considered a study limitation.

Taking into account that the aim of the study was to evaluate acute hormonal disturbances induced by low-dose-OP exposure, no histological changes were expected. 

## 5. Conclusions

Organophosphate intoxication causes a variety of pathophysiological alterations both through the classical mechanism of action represented by the inhibition of acetylcholinesterase and through the multitude of non-cholinergic effects recently described. Currently, the number of cases of severe acute poisoning has significantly decreased; however, due to the extensive use of this type of insecticide, the exposure has become ubiquitous. If past studies dedicated to this topic had as the main objective the description of muscarinic and nicotinic manifestations, nowadays, the neurotoxic, metabolic, and endocrine effects of OP exposure are the main topics brought to attention. In the present research, the obtained results support data on the toxic effects of chlorpyrifos on thyroid function. However, the study has found a particularity in terms of the initial increased level of T3, given that euthyroid thyroid damage is characterized by a rapid decrease in T3 levels. Elevated TSH levels two hours after intoxication are suggestive of possible transient hypothyroid status. Statistically significant correlations, however, support a physiological limitation of the ability to regulate thyroid function. Clearly, given the conflicting data currently available in the literature, as well as the complex toxic effects of OP compounds, this topic requires further research to establish the alterations that can follow acute intoxication.

## Figures and Tables

**Figure 1 toxics-10-00511-f001:**
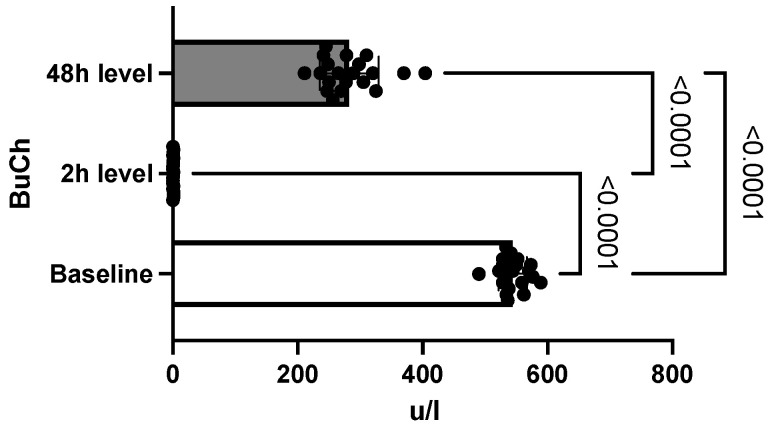
Differences between butyrylcholinesterase (BuCh) levels. (Baseline is defined as a non-OP exposed status. Bars in the graphic show a marked inhibition of BuCh activity after OP exposure, with increasing value at 48 h. BuCh variation confirms the OP intoxication.).

**Figure 2 toxics-10-00511-f002:**
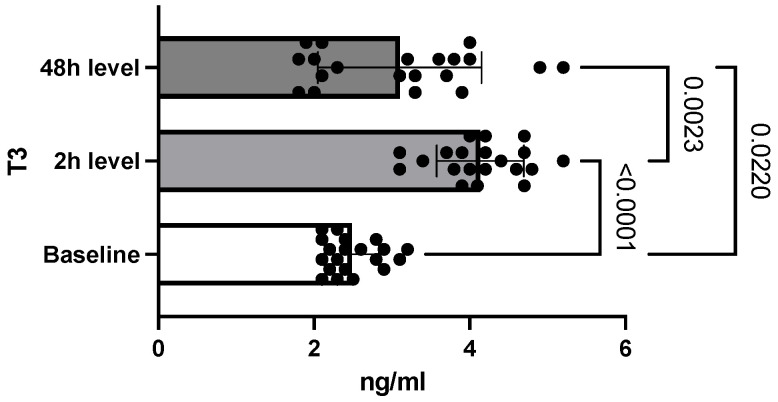
Differences between T3 levels. (Bars display the variation of T3 levels in the study group, from a determined baseline level, with significant increase at 2 h and hormone decrease at 48 h after intoxication.)

**Figure 3 toxics-10-00511-f003:**
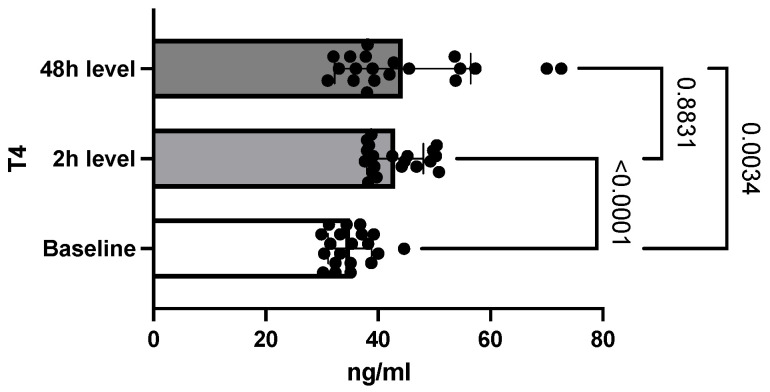
Differences between T4 levels. (Bars display the variation of T4 levels in the study group, with continuous increase at 2 h and 48 h after intoxication, compared to the non-exposed population at baseline.)

**Figure 4 toxics-10-00511-f004:**
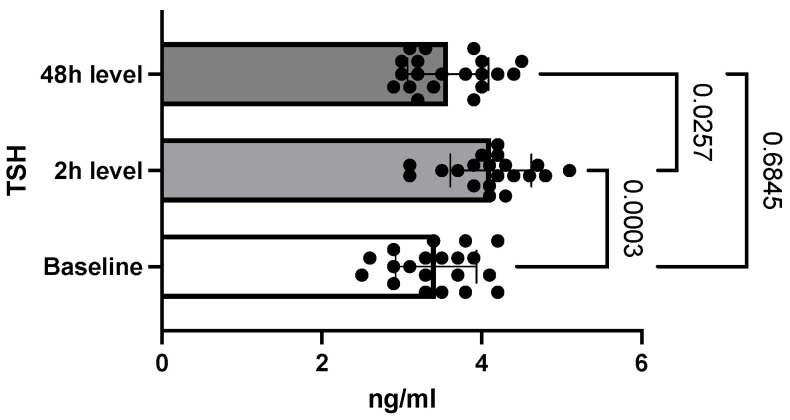
Differences between TSH levels. (Graphic reveals that TSH first increased at 2 h compared to baseline levels and later decreased at 48 h.)

**Figure 5 toxics-10-00511-f005:**
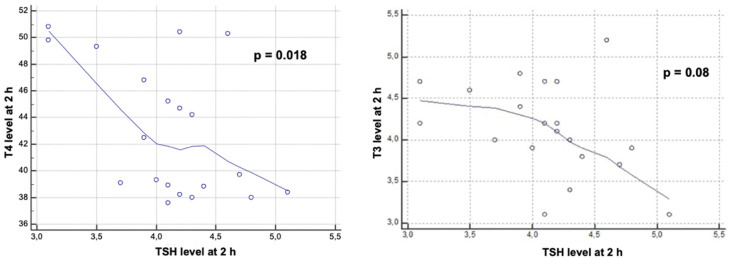
Correlation between T3 (right) and T4 (left) and TSH levels at 2 h. (Scatter plots show that the increases in T4 and TSH at 2 h post intoxication are inversely proportional, and so is the correlation between T3 and TSH).

## Data Availability

The data presented in this study are available on request from the corresponding author.

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
