# Peer review of "Toxic-Induced Nonthyroidal Illness Syndrome Induced by Acute Low-Dose Pesticides Exposure—Preliminary In Vivo Study"

_toxics, 2022, doi:10.3390/toxics10090511_

Round 1

Reviewer 1 Report (Previous Reviewer 2)

All the comments and suggested changes have been adressed.

Author Response

Dear Reviewer,

Thank you very much for your time used to read my paper. It is really very important to me considering the amount of work.

Reviewer 2 Report (Previous Reviewer 1)

I think that we cannot understand this information without a reference, control group

Author Response

Dear Reviewer,

Thank you for your time used to read my work.

I appreciate your comment, however, this research is already finished and  no new data from a control group may be obtained. Moreover I also think this will also be perhaps unethical. 

Considering that this study used a within-subjects design to evaluate the effects of OP exposure, in order to increase internal validity some measures were taken to limit the impact of various extraneous factors which may interfere with the final results, such as sampling at the same time of the day, same light and heat exposure.

Nevertheless, I assumed the lack of the control group in the section for study limits.

Thank you 

This manuscript is a resubmission of an earlier submission. The following is a list of the peer review reports and author responses from that submission.

Round 1

Reviewer 1 Report

There are several things to fix throughout the manuscript (Figures, statistics description, ...). However, I think it must be rejected because of two main reasons:

  1. AChE levels were not analyzed, and this is central here.
  2. Control group is essential, and there is not. With this experimental desing, we have no idea whether everyting around the exposure (gavage, stress, vehicle...) is influencing your results or not.

The first point could be fixed as this sort of analysis could be done in one/two days, but the second point requires doing the whole experiment again (not only a new cohort including only the control group, as cohorts should be composed of around 50% animals of each condition).

Reviewer 2 Report

This is an interesting paper addressing the thyroid hormonal status after exposure to the organophosphate pesticide chlorpyrifos in Wistar rats. The manuscript is well structured, the objectives are clear, the methodology is adequate and has interesting results and discussion. However, I have some considerations and questions before its final consideration for publication: 

1) Keywords: please change pollutans for pollutants

2) Line 24: change acetyl cholinesterase for acetylcholinesterase, as far as in the rest of the text it is referred this way (without separation).

3) Line 26: change the part of the text where you refer to adult male Wister rats to Wistar rats.

4) Line 27: In the abstract, the dose of chlorpyrifos is stated as 0.1mg/kg but then in the material and methods is described as chlorpyrifos 0.1g/kg. Please correct the dose in the text.  

5) Line 30: In the abstract, the authors say […] confirmed major inhibition immediately after intoxication compared to the control group. Compared to the control group or compared to baseline?

6) Line 92: Regarding the experimental design, it would be convenient to include a control group. Specially because there are several factors that may be modifying the final determinations, for instance: animal manipulation, the use of anesthetics for the sample obtention and the use of corn oil for the administration of chlorpyrifos. For this reason, a control group administered with the vehicle should be included. Otherwise, the authors may consider including this to the study limitations.

7) Line 94: In the materials and methods, the authors justify that only male rats were included in order to avoid any hormonal interferences. Taking into account that previous studies with chlorpyrifos found sex-dependent effects, have the authors considered including both males and females? The authors justify the decision referring to hormonal interferences in females during the estrous cycle. Could the authors please provide some references? The authors may consider including this to the study limitations.

8) Line 106: Change chlorpyrofos for chlorpyrifos.

9) Line 108: The link provided in the text is not working www.who.int/whopes/quality/Chlorpyrifos

10) Line 109: In the text, the authors say that the total dose administered to each subject is shown in Table 1. Table 1 is not present in this version of the manuscript.

11) Line 116: In the materials and methods, the authors say that all laboratory animals selected for the study were fed with specific food. Could the authors specify which specific food were the animals fed? If they were not fed with regular rodent chow food, specify the content and the reference of the chosen one, and justify the convenience of using this specific food.

12) Line 135: Regarding statistics, only the differences between the three determinations are presented in the results section. The differences between the baseline and each determination should be included (baseline vs 2h; baseline vs 48h). This is especially important because in the discussion section the authors discuss each of the results independently, and we cannot know whether the observed differences they are statistically significant or not. For exemple:

  • Two hours after exposure, an increase in T4 values was observed, and measurements at 48 hours indicated a continuing upward trend (line 248).

13) Line 149: A general comment about the figures in this manuscript would be to use the same letter format and letter size within each figure. Besides, in the TSH figure, the lowest concentrations is not visible.

14) Line 208-210: Please revise the sentence About two hours after administration, three subjects presented with muscarinic syndrome exhibiting manifestations such as bronchorrhea, polyuria, sialorrhea, and diarrhea, even though the whole group received a dose equivalent to body weight.

15)Line 237 – 247: Revise this part of the discussion regarding T3 levels, specifically the one referring to the article from Güven et al. (1999).

16) References: the authors should use the same format when referring to a specific publication in the text. Please revise all the references in the manuscript. Examples:

Satar reported in his experimental study […]; The study of Wang et al. stands as […]; A recent study published by Leemans et al showed […]; In Slotkin's study, T4 levels were […].

17) Discussion: There are several recent studies analyzing the effects of CPF exposure to thyroid hormones in rats which could be useful for the discussion.

Reviewer 3 Report

The authors of the paper entitled: Toxic-induced Nonthyroidal Illness Syndrome induced by Acute Low-dose Pesticides Exposure – preliminary in vivo study aimed to identify the potential effects of acetylcholinesterase inhibition mediated by chlorpyrifos exposure on thyroid hormones. The introduction is well written and gives a sufficient level of published knowledge on this topic, the results are well written and discussed.

However, I have some major concerns about the methodological plan:

  1. The authors should indicate the location where they buy the animals, as well as the iodine content of the diet.
  2. Did the authors include a control group of animals in the experiment?
  3. The authors should clarify the exposure condition, for how long the animals have been exposed to the chemical?
  4. The biochemical evaluation of thyroid hormones could not be sufficient and exhaustive to comprehend the toxic effects of chlorpyrifos on thyroid tissue, why the authors did not perform other techniques (e.g. histopathology at least)?